# Chest X-ray in Emergency Radiology: What Artificial Intelligence Applications Are Available?

**DOI:** 10.3390/diagnostics13020216

**Published:** 2023-01-06

**Authors:** Giovanni Irmici, Maurizio Cè, Elena Caloro, Natallia Khenkina, Gianmarco Della Pepa, Velio Ascenti, Carlo Martinenghi, Sergio Papa, Giancarlo Oliva, Michaela Cellina

**Affiliations:** 1Postgraduation School in Radiodiagnostics, Università degli Studi di Milano, Via Festa del Perdono, 7, 20122 Milan, Italy; 2Radiology Department, San Raffaele Hospital, Via Olgettina 60, 20132 Milan, Italy; 3Unit of Diagnostic Imaging and Stereotactic Radiosurgery, Centro Diagnostico Italiano, Via Saint Bon 20, 20147 Milan, Italy; 4Radiology Department, Fatebenefratelli Hospital, ASST Fatebenefratelli Sacco, Milano, Piazza Principessa Clotilde 3, 20121 Milan, Italy

**Keywords:** artificial intelligence, chest X-ray, emergency radiology, deep learning, chest radiography

## Abstract

Due to its widespread availability, low cost, feasibility at the patient’s bedside and accessibility even in low-resource settings, chest X-ray is one of the most requested examinations in radiology departments. Whilst it provides essential information on thoracic pathology, it can be difficult to interpret and is prone to diagnostic errors, particularly in the emergency setting. The increasing availability of large chest X-ray datasets has allowed the development of reliable Artificial Intelligence (AI) tools to help radiologists in everyday clinical practice. AI integration into the diagnostic workflow would benefit patients, radiologists, and healthcare systems in terms of improved and standardized reporting accuracy, quicker diagnosis, more efficient management, and appropriateness of the therapy. This review article aims to provide an overview of the applications of AI for chest X-rays in the emergency setting, emphasizing the detection and evaluation of pneumothorax, pneumonia, heart failure, and pleural effusion.

## 1. Introduction

Over recent years, there has been increasing interest in the application of artificial intelligence (AI) techniques to medical imaging examinations. Chest X-ray (CXR) is one of the most frequently performed examinations, particularly in the emergency setting, due to its widespread availability, low costs, and the possibility to be performed at the patient’s bed. It provides significant information on lung parenchyma and the related pathologies, as well as on cardiovascular circulation and pleural disorders.

A correct and rapid CXR report is decisive in choosing the proper treatment and improving the patients’ outcome. Although CXR reading is considered a basic radiological skill, it remains challenging and depends on the radiologist’s experience, workload, and environment. The development of robust and efficient AI algorithms could greatly facilitate CXR readout, particularly in emergency settings, where time spent on CXR interpretation and the accuracy of responses is often of vital importance.

Owing to the high number of CXR examinations performed daily in hospitals around the world, there is a large amount of data available for developing robust AI algorithms. AI-based tools have been shown to facilitate the detection of numerous health-threatening conditions, as well as to prioritize the reporting of patients with critical findings [1,2]. In this narrative review, we provide an overview of the different AI tools used for CXR interpretation and their performance in the emergency setting for the study of: pneumothorax, pneumonia, COVID-19, heart failure and pleural effusion.

### 1.1. A Quick Introduction to AI

Artificial intelligence (AI) can be defined as technology that mimics human cognitive processes, such as learning, reasoning, and problem-solving. As, in conventional radiology, diagnosis is primarily qualitative, AI-powered assessment could make a significant contribution in this field, reducing the variability in image interpretation and improving diagnostic accuracy [1].

AI applications in radiology are driven by the idea that medical images are a set of data that can be computed by a machine to extract useful information [3]. Therefore, increasing the data storage capacity and computing power is a prerequisite for the development of AI-based tools. However, it is not just a question of making the evaluations routinely expressed by the radiologist faster and more precise, but also of extracting information not visible to the human eye to improve the clinical managment [4]. The systematic application of a quantitative approach (essentially AI-driven) to the problem of interpreting biomedical images is at the base of the so-called radiomics paradigm [4,5].

Machine learning (ML) is a branch of AI that applies concepts and tools from other disciplines, primarily statistics and programming, to build algorithms aimed at the automated detection of meaningful patterns in data, a field closely related to data mining [6]. In radiology, ML can be used to extract information from imaging data [4].

The process of developing ML-based tools embodies several phases, among which the main ones are the training and validation phase [6]. In general, the training phase requires the exposure to a set of data, or cases, which can be variably labeled (supervised learning) or unlabeled (unsupervised learning) [6].

In supervised learning, the simplest form of ML, the system simulates the human cognitive process of “learning by examples”. This type of ML is suitable for very general classification tasks in which new elements need to be labeled according to some predefined categories [7]. Labeled data points might be obtained from human experts that annotate (“label”) data with their corresponding label values (for example, a chest X-ray could be positive for pneumonia “yes = 1” or negative “no = 0”). These methods exploit a training set that consists of tuples (x,y) made of inputs (x), for which we know the corresponding label values, which therefore represent the output (y). The supervised ML algorithm searches for a hypothesis (f) that maps the relationship (x → y) between the data, imitating the human annotator, which allows it to predict the label solely based on the features of a data point.

While radiologists mainly evaluate qualitative features, such as increased or reduced radiopacity, comparing them to a subjective reference standard, ML features are low-level properties, or metrics, of a data point that can be computed or measured easily. There are hand-crafted features (manually defined by data scientists) and automatically extracted features (usually through deep-learning algorithms, see further). The problem of choosing which features to select to build more accurate models is one of the most challenging parts in the radiomic workflow. Even if reproducible features are not necessarily clinically informative, successful AI models must be built upon reproducible and robust features [8].

In unsupervised learning, the system analyzes and extracts significant features from unlabeled data by forming groups or identifying relationships between subgroups [6]. This type of ML is suitable for clustering or associative tasks [7].

At the basis of each ML approach are models that can be trained and tested in data analysis [6]. A model is a theoretical hypothesis that maps a possible relationship between data and it is usually based on statistical assumptions that are computationally feasible, meaning that they can be translated easily into the programming code to perform automated data analysis [6]. Through automation, it is possible to test the feasibility of a model for a certain dataset and compare the performances of different models [6]. Several mapping hypotheses can be used to infer predictions of an amount of interest that satisfies some predefined requirement, as in supervised learning, or to ensure some rules of internal consistency between clusters, as in unsupervised learning. Some models may outperform the others in representing the desired relationship, leading to more accurate predictions. The testing phase, which follows the training phase, is necessary to assess the fitness of the model in mapping the desired relationship (if existing) between the data [4].

Artificial neural networks are a particular learning paradigm inspired by the biological network of the human brain [9]. Their operating principle is not based on statistical hypotheses generated a priori, but on the peculiarities of their structure and on computational properties of the units that compose them. In an artificial neural network, each node represents a cell that operates on the input information, according to certain rules, to obtain an output that it transmitted to the next neuron to be further processed. The computation performed by a single unit is influenced by its interconnections and their weight, which provide a measure of how much each input “counts” in the neuron. In this way, the flow of information is passed through the network while shaping the network itself. The global function (for example image recognition) is obtained through the coordinated activity of smaller units each performing an elementary computational function [9].

Complex artificial neural networks, called Convolutional Neural Networks (CNNs), have been developed and found to be particularly suitable for image analysis and recognition tasks. Deep learning (DL) is a domain of AI that takes advantage of complex artificial neural networks such as CNNs to discover intricate patterns in data. DL networks feature many intermediate layers, where each layer represents increasing levels of abstraction, to the extent that it is unclear exactly how processing the intermediate layers contributes to the overall result [10]. This is also known as the black box phenomenon and contributes to the problem of interpretability of the results of AI tools.

DL models are built to capture the full image context and learn the correlations between the local features, resulting in a superior performance in various radiological tasks, such as interpreting radiographic exams.

### 1.2. Open Datasets

ML and DL algorithms are trained with datasets and are dependent on the number and quality of the training data. There is a constantly increasing number of publicly available CXR datasets that can be used for image classification and retrieval tasks. Some of the biggest and most commonly used open datasets are listed in Table 1.

## 2. Worklist Prioritization

The automatic notification of critical findings is one of the most interesting AI applications in emergency radiology. With the increased demand for imaging studies, delaying the communication of key data to the treating physician can delay critical care and compromise therapeutic efficacy, particularly in urgent scenarios [21].

The priority assigned by the emergency doctor who first examines the patient determines the sequence in which the imaging exams are reported; unfortunately, the precedence is not always consistent with the abnormalities observed. AI-based models may detect and prioritize emergency CXR findings in real time, reduce the report response times for key findings, and optimize therapeutic pathways.

A notification system developed by GE Medical System and Zebra Medical Vision for the evaluation of pneumothorax on CXR demonstrated a significant reduction in the time required for the diagnosis. Three experienced radiologists evaluated 588 CXR with the HealthPNX prioritization software with an average diagnosis time of 8.05 min versus 68.98 min without the software. The time needed to assess the radiograph and send a notification was only 22.1 s [22].

Annarumma et al. created and tested a CNN-based tool to simulate an automatic triage for adult CXRs according to the urgency of the imaging findings. The use of the algorithms resulted in a theoretical reduction in the reporting delay for critical studies, from 11.2 to 2.7 days [23]. Another AI tool developed by Kim et al. allowed the reduction of the time-to-report for CXRs of critical and urgent cases (from 3371.0 to 640.5 s and from 2127.1 to 1840.3 s, respectively) [24].

In the setting of the COVID-19 pandemic, Tricarico et al. [25] developed an automated tool for the prioritization of patients with the suspicion of the COVID-19 disease based on CXR analysis. This CNN-based system aimed to facilitate the workload in the emergency department by fast-forwarding the testing of suspicious cases. The proposed architecture was reviewed retrospectively on a dataset of cases collected throughout the first months of the pandemic and showed significant improvements for the identification and prioritization of COVID-19 patients. The system’s sensitivity and specificity were 78.23% and 64.2%, respectively. In preliminary real-life testing, the method reached a correlation of 0.873.

## 3. Pneumothorax

Pneumothorax is a pathological condition in which the pleural cavity fills with air, impairing oxygenation and ventilation. It occurs spontaneously or as a complication of trauma, medical interventions, and infections. Due to the sheer variety of the underlying etiologies and clinical scenarios, pneumothorax represents an important morbidity and mortality factor. Some forms present with a severe progressive hemodynamic compromise, with eventual cardiovascular collapse and respiratory failure if left untreated.

CXR allows for the timely diagnosis and objective quantifying of pneumothorax, which is crucial for the selection of the optimal management strategy. AI could potentially increase the sensitivity for pneumothorax identification and provide quantification through volume segmentation, particularly in low-resource settings where experienced radiologists might be lacking. Automated pneumothorax detection represents a challenging technical task due to the variability of its appearances on CXR. However, a variety of AI solutions have been proposed in recent years, powered by the rapid development of DL and the availability of large CXR datasets.

In 2017, Wang et al. published a large ChestX-ray8 database with image-level labels for eight chest conditions, including pneumothorax. A multilabel deep CNN model was performed on the dataset and demonstrated an accuracy of 0.0816 for pneumothorax detection, with an average false positive rate of only 0.2317 [26]. Smaller datasets have also been used for the training of DL algorithms. For example, Blumenfeld et al. used a dataset of 117 CXRs with pixel classification and reached a diagnostic accuracy of 0.95 for pneumothorax detection [27].

A model involving CNNs on frontal CXR demonstrated a sensitivity of 0.55, a specificity of 0.90, and an area under the curve (AUC) of 0.82 for the assessment of large and moderate pneumothorax on internal testing, although the performance was lower on external testing (AUC = 0.75) [28].

In a pneumothorax segmentation competition organized in 2019 by the Society for Imaging Informatics in Medicine, the winning team achieved a Dice score of 0.8679 by using a deep neural network ensemble and extensive data pre-processing and augmentation [29]. Wang et al. proposed a construct of several modified U-Net convolutional network models, which were validated at the 2019 segmentation competition and reached an area under the curve of 0.9795 and a Dice score of 0.8883 [30]. Another model based on the U-Net CNN architecture showed an accuracy of 97.8% and sensitivity of 69.2%, which was less precise compared to that of an experienced radiologist, but did not differ significantly (*p* = 0.11). For patients with >21.6% of pneumothorax, the model predicted the need for thoracostomy [31].

In another study, using a fully convolutional network algorithm trained on a large dataset with pixel-level labels, the authors reached 93.45% diagnostic accuracy and high segmentation accuracy with a mean pixel-wise accuracy (MPA) of 0.93 ± 0.13 and dice similarity coefficient of 0.92 ± 0.14 [32].

Several authors have proposed pneumothorax detection algorithms based on ResNet artificial neural networks. Gooßen et al. demonstrated an AUC of 0.96 for a ResNet-50 model [33].

A Deep ResNet-50 model successfully detected pneumothorax with a combined Dice score of 0.82 and allowed for the segmentation of pneumothorax lesions, with a Dice score ranging between 0.72 and 0.79 [34]. In a recent study, two-stage ResNet algorithms trained and validated on a large dataset demonstrated an accuracy of 94.4% and an area under the curve of 97.3% for the detection of pneumothorax [35].

Another CNN-based model correctly recognized pneumothorax and tension pneumothorax cases with an AUC of 0.979 and 0.987, respectively [36].

Yi et al. compared the performance of an algorithm based on ResNet-152 deep CNN with that of first-year radiology residents. Although the model performed faster than the first-year radiology residents, with 1980 and 2 images assessed per minute, respectively, its AUC was significantly lower (0.841 vs. 0.942 and 0.905 (*p* < 0.01)). The deep CNN identified 9.7% of the pneumothoraxes missed by at least one of the residents [37]. While DL algorithms are not sufficiently robust to independently assess CXR for pneumothorax, recent developments have highlighted their potential role as supportive tools. A multicenter cohort study demonstrated that the AI-aided interpretation of CXR by radiologists showed significant improvement of AUROC for pneumothorax [38].

## 4. Pneumonia

Pneumonia constitutes a major health hazard despite the advances in its diagnosis and management. A variety of agents can give rise to pneumonia, which translates into the heterogeneity of its epidemiology, signs and symptoms, presentation on diagnostic tests, and clinical course. Pediatric pneumonia is an ongoing global healthcare challenge, which accounts for 14% of all deaths of children under five years old, according to the World Health Organization.

CXR is the first imaging test performed for pneumonia diagnosis. Moreover, the correct interpretation of CXR allows for differentiating between viral and bacterial etiology of pneumonia, with added value for patient management. This is particularly important in developing countries, which account for a large percentage of childhood morbidity and mortality from pneumonia, but have limited access to other diagnostic tests. Not surprisingly, the diagnosis of pneumonia via CXR, particularly in the pediatric setting, has attracted significant interest among AI researchers, with a variety of proposed models.

The multilabel ChestX-ray8 project, which was described before, demonstrated an accuracy of 0.75 for the detection of pneumonia, with an average false positive rate of 0.0691 [26]. A transferable CNN, which demonstrated efficiency in classifying age-related macular degeneration and diabetic macular edema, showed an accuracy of 92.8% and an AUC of 96.8% when applied and trained on the pediatric CXR dataset for pneumonia detection. Moreover, the model was able to reliably differentiate between viral and bacterial pneumonia (accuracy 90.7%, AUC 94.0%) [39].

Similarly, a customized VGG16 model reached 96.2% diagnostic accuracy and 93.6% classification accuracy for distinguishing between bacterial and viral pneumonia, respectively. It also integrated a novel strategy for visualizing the algorithm region of interest on CXR for improved transparency of deep learning inner workings and behavior [40].

Gu et al. developed a full CNN for the segmentation of the lung regions followed by deep CNN for classification, which was evaluated on an internal pediatric CXR dataset with an accuracy of 0.80 and sensitivity of 0.77 [41]. A CNN model by Okeke et al. showed an accuracy of 0.93 for the detection of pneumonia on pediatric CXR [42]. Another CNN reached a similar accuracy, of 96-97%, in the Kaggle pneumonia dataset [43]. A CNN model by Liang et al. showed 96.7% accuracy for the detection of pneumonia in pediatric patients [44].

Rahman et al. assessed four pre-trained CNN models (AlexNet, ResNet18, DenseNet201, and SqueezeNet) on a large CXR dataset and demonstrated the superior performance of DenseNet201, which detected pneumonia with an accuracy of 0.98 and differentiated between viral and bacterial etiologies with an accuracy of 0.95 [45]. In a study by Toğaçar et al., a CNN algorithm with a linear discriminant analysis feature yielded an accuracy of 99.41% for the detection of pneumonia on CXR [46], whereas the novel deep separable residual learning model reached 98.8% accuracy and 0.99 AUC values [47].

A systematic review and meta-analysis by Li et al. demonstrated pooled AUC of 0.99 (95% CI: 0.98–100) across 15 studies assessing the performance of DL algorithms for pneumonia detection on CXR. The pooled sensitivity and specificity were 0.98 and 0.94, respectively. Moreover, it showed a pooled AUC of 0.95 for differentiating between bacterial and viral pneumonia on CXR. However, it was noted that the included studies lacked performance in comparison with healthcare professionals [48].

In a recent study by Kwon et al., an ensemble CNN model demonstrated an AUC of 0.983 for detecting pneumonia on CXR and showed a predictive value in differentiating cases that were improving and those that worsened over seven days of follow-up (*p* = 0.001), highlighting the potential role of AI in directing management strategies as well as refining diagnosis [49].

## 5. COVID-19

The outbreak of the COVID-19 pandemic gave rise to a global race toward the development of reliable and accurate diagnostic tools [50]. CXR has been one of the first tools extensively used to screen patients for COVID-19 pneumonia considering its wide availability, considerable prognostic value, and low costs [51,52]. However, interpreting CXR in a COVID-19 setting can be challenging due to its indistinct radiological characteristics, which include consolidation and hazy increased opacities [53]. AI demonstrated the potential to assist radiologists in differentiating COVID-19-positive cases on CXR. The initial lack of wide datasets has been one of the major obstacles to building AI-driven disease detection models. Additionally, CNN training takes a substantial amount of time because of the computational demands and memory constraints. Transfer learning offers an alternative development method that can overcome this issue by using pre-trained models. Utilizing pre-trained networks, such as InceptionV3, VGGNet, InceptionResNetV2, ResNet, etc., the CXR-based identification and detection of COVID-19 has been developed with benchmark accuracies reaching 99% [54].

Baltazar et al. conducted a retrospective clinical analysis on 1171 clinically verified CXR pictures from 821 cohorts that were then made accessible in open-access repositories. Among the five optimized DL architectures, InceptionV3 demonstrated the best performance for COVID-19 pneumonia detection with 86% sensitivity, 99% specificity, 91% positive predictive value, and an AUC of 0.99 in differentiating COVID-19 from negative CXR [55].

Nillmani et al. propose 16 types of segmentation-based classification deep learning-based systems for the automatic detection of COVID-19. The best performing segmentation-based classification model was UNet+Xception, which exhibited the accuracy, precision, recall, F1-score, and AUC of 97.45%, 97.46%, 97.45%, 97.43%, and 0.998 (*p*-value < 0.0001), respectively [56].

A classifier ensemble strategy using the Choquet fuzzy integral was suggested by Dey et al. It divides CXR scans into three categories: confirmed COVID-19, common pneumonia, and healthy lungs. Using two dense layers and one softmax layer, they extracted characteristics from the CXR pictures and classified them using the pre-trained convolutional neural network models. The accuracy provided by the suggested approach is 99% [57].

The approach presented by Nasiri et al. combines the MobileNet and DenseNet169 Deep Neural Networks to extract the characteristics of patient’s X-ray images. They subsequently used the chosen characteristics as input to the classification algorithm LightGBM (Light Gradient Boosting Machine). The ChestX-ray8 dataset, which comprises 1125 X-ray images, was used to evaluate the performance of the suggested approach. In the two-class (COVID-19, Healthy) and multi-class (COVID-19, Healthy, Pneumonia) classification tasks, they respectively achieved accuracies of 98.54% and 91.11% [58].

Ezzoddin et al. proposed a similar approach using DenseNet169 to extract the features of the patients’ CXR images and LightGBM algorithm in order to classify them. The evaluation of the ChestX-ray8 dataset reached accuracies of 99.20% and 94.22% in the two-class (i.e., COVID-19 and No-findings) and multi-class (i.e., COVID-19, Pneumonia, and No-findings) [59].

In two other studies, Nasiri et al. employed an approach using the DenseNet169 Deep Neural Network (DNN) to extract the features of X-ray images taken from the patients’ chests. In one of the studies, the features were chosen by a feature selection method, i.e., analysis of variance (ANOVA), to reduce the computation and time complexity while overcoming the curse of dimensionality to improve the accuracy. Finally, the extracted features were given as input to the Extreme Gradient Boosting (XGBoost) algorithm in order to perform the classification task. The experiments showed an accuracy of 98.72% and 98.23% for two-class classification (COVID-19, No-findings) and 89.7% to 92% accuracy for multiclass classification (COVID-19, No-findings, and Pneumonia) [60,61].

Other studies have investigated the possibility of determining and classifying images according to the estimated degree of disease severity [62].

Cohen et al. used a DenseNet model to generate a severity score based on CXR imaging. Images from a public COVID-19 database were scored retrospectively in terms of the extent of lung involvement and the degree of opacity. A neural network model that was pre-trained on large (non-COVID-19) CXR datasets was used to construct features for COVID-19 images. As a result, training a regression model on a subset of the outputs from this pre-trained model predicted the geographic extent score with a 1.14 and the lung opacity score with a 0.78 mean absolute error, respectively. The model allowed us to assess the severity of COVID-19 lung infections, which can be used for tracking the effectiveness of the treatment and for escalating or de-escalating care [63]. Jiao et al. evaluated the ability to predict the severity of COVID-19 disease utilizing the CXR as input to an EfficientNet deep neural network together with clinical data. They reported that when CXR was added to clinical data for severity prediction, the AUC increased from 0.82 to 0.84 on internal testing and from 0.73 to 0.79 on external testing. When deep-learning features were added to the clinical data for the progression prediction, the concordance index increased from 0.76 to 0.80 on internal testing and from 0.70 to 0.75 on external testing, concluding that data inferred from CXR through AI applications can augment clinical data in predicting the risk of progression to critical illness in patients with COVID-19 [64].

Khan et al. developed a new CNN architecture, STM-RENet, to interpret radiographic patterns from X-ray images. They proposed a new convolutional block STM that implements the region and edge-based operations both separately and jointly. The learning capacity of STM-RENet was further enhanced by developing a new CB-STM-RENet that exploited channel boosting and learned textural variations to effectively screen the X-ray images. The suggested model performed significantly better than typical CNNs on three datasets, particularly the CoV-NonCoV-15k dataset, with a high detection rate (97%), and accuracy (96.53%) [65].

It is also worthwhile to mention some of the real-world applications of AI systems: Mustaq et al. [66] compared the performance of a deep learning AI-based system (qXR v2.1 c2, Qure.ai Technologies) to the RALE score, a radiographic score with strong inter-observer agreement that has been validated to quantify the severity and prediction outcomes in ARDS patients (Figure 1) [67]. This algorithm was initially created for use on TB patients and Mustaq et al. applied it to 694 patients, concluding that a Qure AI score of ≥30 or a RALE score of ≥12 on the CXR at the emergency department presentation was independent of, and equivalent to, the predictors of bad outcomes. Hasani et al. [68] applied an Automated Detection System utilizing X-ray Images (COV-ADSX), which detects COVID-19 using a deep neural network and XGBoost. The accuracy of the model utilized in COV-ADSX based on the ChestX-ray8 dataset was 98.23% in only 10 s, allowing the expert to receive a response rapidly while waiting for the PCR result.

By assisting in difficult decision-making, AI has been proven to have the potential to play a pivotal role in the fight against COVID-19. AI algorithms can be trained to automatically detect and classify various features of the disease on CXRs with high accuracy. This could potentially save time and resources in the demanding pandemic setting. There are still some challenges that need to be addressed before AI can be widely used for COVID-19 diagnosis. The principal studies are summarized in Table 2.

## 6. Heart Failure

In the context of an increasingly aged patient population, heart failure is one of the major causes of admission to the emergency department. Once again, CXR is one of the first-line tools in the assessment of patients in this setting [69].

Cardiopathic patients may be not compliant and are often bedridden. As a result, CXR is often limited to anterior-posterior projection with the consequent projective enlargement of the heart. In addition, their thorax may present multiple devices, such as electrocardiogram electrodes, pacemakers, or implantable cardioverter-defibrillator, which overlap the lung fields and increase the possibility of erroneous findings [70]. In this context, CXR reporting can become challenging, and a supportive tool could be useful. The Cardio-Thoracic Ratio (CTR) is the most evaluated parameter in these patients, referred to as the ratio between the maximum transverse cardiac diameter and the maximum horizontal thoracic diameter, with cardiomegaly defined as a value higher than 0.50 [71]. An automatic CTR measurement could be a simple and helpful tool to accelerate routine workflow [72]. Precise segmentation is the first step needed for an automatic DL-based calculation of CTR. CardioNet is an automatic segmentation model based on the CNN architecture that can recognize and segment the lungs and heart and generate an intuitive map from CXR, of varying quality, with an accuracy of 98.9%. CardioNet has been trained on 248 images augmented through cropping, horizontal flipping, and translating the initially available images.

A recent clinical evaluation has been performed to compare four DL models (AlbuNet, SegNet, VGG-11, and VGG-16) to find which one would show the highest percentage of automatic results of the cardiac size accepted by users with a measurement variation within ±1.8% of the human-operating range. VGG-16 gave the highest-grade result (68.9%), but the combined AlbuNet + VGG-11 model yielded excellent grades in 77.8% of the images in the evaluation dataset, a coefficient of variation of 1.55%, and reduced the CTR measurement time by almost ten-fold (1.07 ± 2.62 s vs. 10.6 ± 1.5 s) compared with manual operation [73]. Alveolar edema is another sign of heart failure, visible as bilateral perihilar lung shadowing, also known as “bat wing opacities” or “butterfly opacities”. This sign is the result of the hemodynamic pulmonary congestion caused by heart failure that brings high pulmonary capillary pressure and the abnormal transfer of fluid from the vascular to the extravascular compartments of the lungs (interstitium and alveolar spaces) [74]. Blood flow circulation alterations and inflammation result in the abnormal accumulation of fluid, respectively, transudate and exudate, seen as lung opacities on CXR. Despite some radiological similarities, these distinct diseases require different treatments, and early diagnosis of cardiogenic edema is therefore the key to improving patient outcomes.

A reliable AI algorithm should be able to distinguish the most common causes of acute respiratory failure in an emergency, mainly congestive edema and pneumonia [75].

Considering that ML is more efficient in the distinction between localized lesions and lesions with global symmetrical patterns [76], unilateral consolidation pneumonia is easier to identify than diffuse bilateral pulmonary edema. On this basis, Liong-Rung et al. have created an ambitious project using deep CNN models to recognize pneumonia and pulmonary edema in CXR images of older patients. Although the presence of medical devices in the training dataset has brought a decreased predictive performance with an accuracy reduction, from 79.1% to 73.4%, they tried to include those images anyway, attempting to post-process them by cropping, unfortunately, without favorable results [75]. However, they deserve credit for experimenting with DL tools on sub-optimal radiograms in complicated circumstances, those in which radiologists would truly appreciate artificial support. Indeed, the study revealed that using images with explicit signs of edema or pneumonia and without the interference of device overlap for training deep learning models can produce accuracy above 80% in differential diagnosis, while an accuracy of approximately 70% has been achieved in the presence of interference. Lastly, normal CXRs of patients without pneumonia or pulmonary edema had an F1 score over 95% [75].

If CNN models could suit CXR feature recognition, the current scientific literature contains a wide selection of encouraging applications that include more features, including regarding heart failure. For example, Cicero et al. investigated five radiological signs—cardiomegaly, pleural effusion, pulmonary edema, pneumothorax, and consolidation—using the GoogLeNet CNN, trained on a total of 35038 images and tested on a set of 2443 radiographs.

The sensitivity, specificity, and AUC, respectively, were 91%, 91%, and 0.962 for pleural effusion, 82%, 82%, and 0.868 for pulmonary edema, 81%, 80%, and 0.875 for cardiomegaly.

The best results were achieved in classifying a study as normal with an overall sensitivity and specificity of 91% and an AUC of 0.964 [77].

Moreover, another CNN, CheXNeXt, was developed a few years ago to detect the presence of 14 different pathologies in chest radiographs, including cardiomegaly and pleural effusion. In 2018, the model was compared with nine expert radiologists and three senior radiology residents. CheXNeXt did not achieve radiologist-level performance on three pathologies (cardiomegaly, emphysema, hiatal hernia), it performed better than radiologists in detecting atelectasis and there were no statistically significant differences in AUCs for the other ten pathologies, including pleural effusion [76].

## 7. Pleural Effusion

Pleural effusion is a medical condition characterized by the pathological accumulation of fluid between the two pleural leaflets. Usually, it is used as a generic term to describe any abnormal accumulation of fluid in the pleural cavity, also because most effusions are diagnosed by CXR, which cannot distinguish between different types of fluids.

Recently, and with increasing interest, the world of diagnostic imaging has embraced the use of artificial intelligence, testing its applications in a wide variety of contexts [78].

A few papers have been published to validate AI tools in the diagnosis and/or quantification of pleural effusion via CXR. Zhou et al. [79] developed and validated a DL system for the detection and semi-quantitative analysis of cardiomegaly, pleural effusion, and pneumothorax. They included two datasets: one for detection and one for segmentation. The first dataset (used for detection) consisted of 2838 CXRs from 2638 patients containing findings positive for cardiomegaly, pneumothorax, and pleural effusion; the second dataset (used for segmentation) was from two publicly available datasets, containing 704 CXRs. Based on the accurate detection and segmentation, semiquantitative indexes were calculated. The detection models achieved high accuracy in detecting cardiomegaly, pneumothorax, and pleural effusion. Moreover, the authors believed that semiquantitative analysis could reduce the work of radiologists, improve the objective accuracy of the quantitative measurement, and be a reasonable option to assist clinical diagnosis.

Huang et al. [80] developed a DL system to quantify the severity of pleural effusion in the CXR of patients with chronic obstructive pulmonary disease (COPD). For this purpose, they used the MIMIC-CXR dataset, dividing the patients within it as “with” or “without” COPD. The label of pleural effusion severity was obtained from the extracted COPD radiology reports and classified into four categories: no effusion, small effusion, moderate effusion, and large effusion. The selected reports were re-tagged by a radiologist without knowing their previous tags as a verification cohort; 15,620 CXRs with clearly marked pleural effusion severity were obtained (no effusion, 5685; small effusion, 4877; moderate effusion, 3657; and large effusion, 1401). The highest accuracy rate of the optimized model was 73.07. The micro-average AUCs of the test and validation cohorts were 0.89 and 0.90, respectively, and their macro-average AUCs were 0.86 and 0.89, respectively.

Niehues et al. [81] created a DL model used specifically for bedside CXRs to detect clinically relevant findings to help emergency and intensive care physicians to focus on patient care, using the reference standards established by computed tomography (CT) and numerous radiologists. They retrospectively collected 18,361 bedside CXRs of patients treated at a level 1 medical center who had undergone a chest CT within 24 h from the CXR. A DL algorithm was developed to identify eight findings on bedside CXRs (cardiac congestion, pleural effusion, air-space opacification, pneumothorax, central venous catheter, thoracic drain, gastric tube, and tracheal tube/cannula). In case of a disagreement between the CXR and CT, human-in-the-loop annotations were used. The AUC for cardiac congestion, pleural effusion, air-space opacification, pneumothorax, central venous catheter, thoracic drain, gastric tube, and tracheal tube/cannula were 0.90, 0.95, 0.85, 0.92, 0.99, 0.99, 0.98, and 0.99, respectively, showing a similar performance to expert radiologists.

## 8. Limits and Future Perspective

AI applications for thoracic disorders have shown promising outcomes in terms of enhancing the existing clinical systems and prognostic prediction reasoning. According to a recent study that compared the opinions of thoracic radiologists and computer scientists, 15.6% of computer scientists thought that the radiologist’s position will be obsolete in 10–20 years [82]; despite this, the stakeholders’ opinion is that this scenario will be unlikely, although there is great emphasis put on radiologists’ education and training in the AI [83].

The use of open datasets had a significant impact on the development of AI algorithms, but the absence of standardized and well-defined criteria also led to the creation of flawed datasets. This needs to be addressed to increase the performance of AI models through high-quality training. As a result, it is critical to understand the limitations of the datasets for their effective use; the key limitations are summarized in Table 3.

Another key aspect would be the integration of AI education into the post-graduation programs for radiology residents. The possibility of increasing performance through personalized learning is the primary motivator for AI-augmented radiological precision education [89]. According to a 2018 poll, 71% of radiologists do not actively use AI. However, 87% of respondents intend to study and 67% are eager to assist in the development and training of such algorithms [90]. Gaube et al. [91] affirm that observers were typically not opposed to preferring AI guidance over human suggestions, showing that when the improvement of radiologists’ performance with AI assistance was lacking, it was likely driven by their confidence in their own opinion rather than their reluctance to trust in the AI-algorithm.

Imaging research is seeing an increase in the number of articles offering novel diagnostic, classification, and prediction tools based on ML that outperform the previous techniques. However, few studies have been conducted to assess the applicability of these models and the tangible advantages they offer to clinical practice in the real-world setting [92]. When such models are evaluated in other patient cohorts or institutions, they demonstrate a lack of reproducibility. Moreover, it is difficult to compare the performance of different ML models. This is due to one of the primary obstacles to effective AI application today: a lack of empirical data confirming the efficacy of AI-based interventions in prospective clinical trials. The existing empirical research is limited and primarily focuses on AI in the general workforce rather than its impact on patient outcomes [93]. Among 516 eligible studies evaluated by Kim et al., only 5% performed external validation and none of these adopted the three design features (diagnostic cohort design, inclusion of multiple institutions, and prospective data collection) [94]. Another critical issue is that most of the existing radiological DL models only incorporate imaging information, without accounting for the background clinical data. We need research on DL models capable of integrating multimodal data, such as clinical and genetic information [95]. The integration of these data has already shown potential for improving prognostic models for other pathologies. For example, Xu et al. used bidirectional deep neural network (BiDNN) DL architecture to combine RNA-Seq and DNA methylation data from a group of gastric cancer patients to identify different prognostic levels [96]. In a more recent study by Cheerla et al., a C-index of 0.78 was found in a pancreatic cancer survival prediction model that used clinical, mRNA, miRNA, and whole-slide imaging data [97]. Emergency departments are a crucial training scenario for determining the benefits and usability of AI-based solutions. In this case, AI assistance might enhance both the performance of a single radiologist and the organization’s management [98].

Last but not the least, the ethical implications of AI integration should be considered before they can see widespread integration into clinical practice. When it comes to AI applications that attempt to enhance patient outcomes, the issue of responsibility becomes considerably more important, particularly when problems and errors occur. At present, it is unclear who should bear responsibility if the system fails. One of the possible solutions includes a joint effort by governments and industries to define the criteria of transparency and accountability, which aspects of the incoming data are influencing the outputs, and utilizing that to deduce what is happening within the “black box” [78]. As described above, an emergency setting is an essential testing ground for the usability and advantages of AI-based solutions. In this scenario, AI systems might assist radiologists with long and highly repetitive activities, minimizing diagnostic mistakes in situations where workload, expectations, and the risk of error are high [98]. Effective AI integration might free up resources that could be committed to other tasks, such as patient communication, that have been overburdened as the workload has increased [99].

## 9. Conclusions

We have provided an overview of AI applications for CXR in the emergency setting.

We believe that AI will transform the healthcare system, notably, in the imaging sector. Radiologists should know the AI technologies that are currently available and use them when appropriate to improve their diagnostic accuracy and treatment planning in the emergency setting.

## Figures and Tables

**Figure 1 diagnostics-13-00216-f001:**
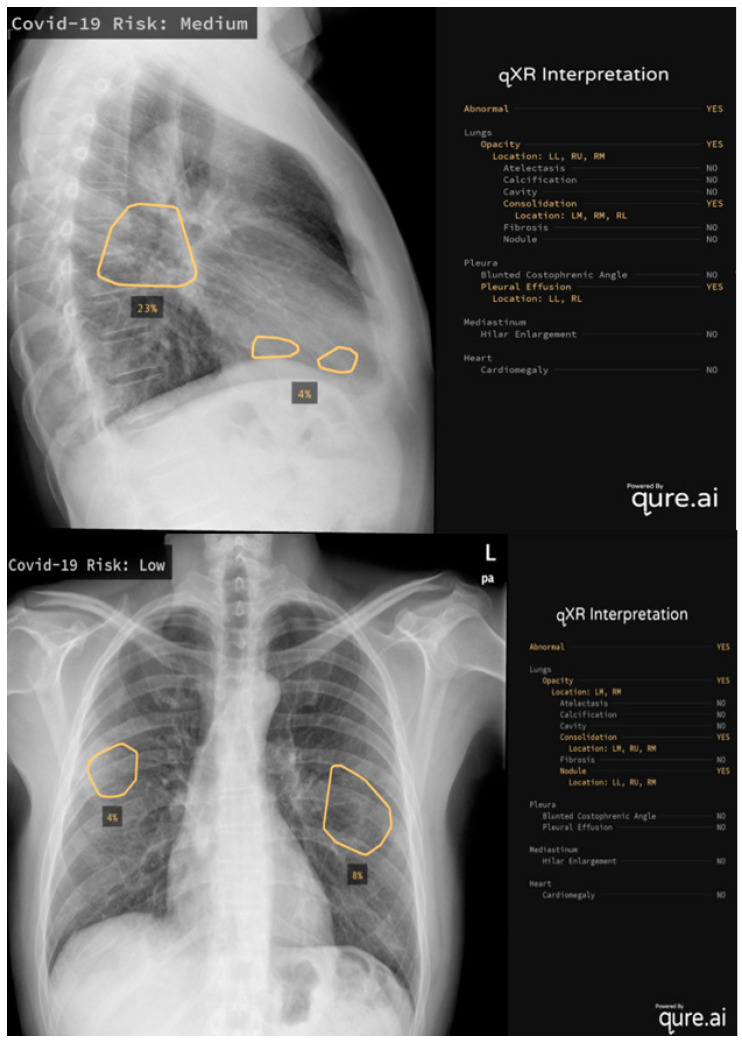
Two examples of qure.ai application, the algorithm calculates the risk levels, evaluating lungs, pleura, mediastinum and heart on CXR.

**Table 1 diagnostics-13-00216-t001:** The table summarizes the characteristics of the main CXR datasets such as the number of images, number of labels, and labeling technique. The dataset containing more images is the CheXpert with 14 labels. In the last two columns, we have listed the main strengths and weaknesses of the different datasets.

Dataset Name	Country	Images	Studies	Labels	Labels’ Technique	Format	Limitations	Strengths
**MIMIC-CXR** [11]	Israel	371,920	65,383	14	Natural Language Processing	JPEG and DICOM	Absence patient demographic data	Number of cases
**CheXpert** [12]	USA	224,316	65,240	14	Natural Language Processing and radiologist consensus on CXR	JPEG	No statistical test to assess the difference between radiologists and the model	Introduction of pre-negation and post-negation stage for classification of uncertainty
**ChestX-ray14** [13]	USA	112,120	30,805	14	NLP and Radiologist interpretation of CXR	PNG	Many findings are not included in radiology reports.	Number of labels
							Labe disambiguation failure (“emphysema” in case of subcutaneous emphysema) [14]	
**PLCO** [15]	USA	185,421	56,071	12	Radiologist interpretation of CXR	TIFF	Chest Xray made for a lung cancer screening program	Number of cases
**PadChest** [16]	Spain	160,868	67,625	193	Natural Language Processing (73%) and Radiologists’ report interpretation (27%)	DICOM	Under-reporting bias: not all the features are listed in the report	Number of labels
							The severity of medical condition is not currently captured in the labels	
							Selection bias including only CXR with available reports	
**BRAX** [17]	Brazil	40,967	19,351	14	NLP, radiologists’ report interpretation and Radiologist interpretation of CXR	DICOM and PNG	Absence of other metadata (gender and race)	1000 reports were randomly reviewed by two radiologists
**Indiana University dataset** [18]	USA	7470	3955	177	Radiologist interpretation of CXR and Natural Language Processing	DICOM	Number of patients	Frontal and lateral CXR and number of features
**Ped-Pneumonia** [19]	USA	5856	5232	2	Radiologist interpretation of CXR	JPEG	Number of features	Pediatric cases
**RSNA Pneumonia** [20]	USA	30,000		1	Radiologist interpretation of CXR	DICOM	Only pneumonia	4527 cases were read by 3 radiologists

**Table 2 diagnostics-13-00216-t002:** Accuracy of most recent studies for COVID-19 diagnosis using artificial intelligence and CXR.

Study	Type of Images	Dataset Used (C-COVID-19, N-Normal, P-Pneumonia)	Method	Accuracy (%)
Baltazar et al. (2021) [55]	Chest X-ray	N:3593	InceptionV3	96
		C:629		
Nillmani et al. (2022) [56]	Chest X-ray		UNet+Xception	97.45
Dey et al. (2021) [57]	Chest X-ray	N:739	Choquet fuzzy integral based ensemble	99
		C:1072		
		P:3100		
Nasiri et al. (2022) [58]	Chest X-ray	ChestX-ray8	MobileNet and DenseNet169+LightGBM	98.54 (two class) 91.11 (multi-class)
Ezzoddin et al. (2022) [59]	Chest X-ray	ChestX-ray8	DenseNet169+LightGBM	99.20 (two class)
				94.22 (multi-class)
Nasiri et al. (2022) [60]	Chest X-ray	ChestX-ray8	DenseNet169+XGBoost	98.23–98.72 (two-class)
				89.7–92 (multi-class)
Cohen et al. (2020) [63]	Chest X-ray	N: 88079	DenseNet	-
		C: 94		
Jiao et al. (2021) [64]	Chest X-ray	N+C: 1834	EfficientNet+Clinical Data	Data inferred from CXR through AI applications can augment clinical data in predicting the risk of progression
Khan et al. (2022) [65]	Chest X-ray	CoV-NonCoV-15k dataset	CB-STM-RENet	96.53

**Table 3 diagnostics-13-00216-t003:** This Table lists the main limitations related to the datasets used to develop AI algorithms.

Datasets Limit	Features
Dataset shift DOI [84]	A primary cause of AI system failure: when a machine-learning system underperforms due to a mismatch between the data set with which it was designed and the data on which it is deployed, this is known as dataset shift.
Annotations [85]	How to annotate the massive volume of medical images required by deep learning models while maintaining quality because of the medical expertise required, large-scale crowd-sourced hand-annotation, such as ImageNet is not viable. Shin et al., for example, developed a model to detect a disease from a CXR training CNNs with 17 distinct illness annotation patterns and controlled vocabulary phrases (Medical Subject Headings (MeSH) to label the different patterns [78].
Significance [86]	The clinically relevant image labels that must be established can sometimes be challenging, in the case of “hedging statement”, for example, is difficult to say if the label is correct when we find in a report “possibly due to emphysema”. The correctness, meaning, and clinical significance of the labels can all be negatively impacted, especially if the dataset generation process is not well described and the labels created are not extensively reviewed. These issues may be minimized by using an experienced visual inspection of the label classes, as well as extensive documentation of the creation process.
Radiologist reports [87,88]	The absence of structured reports makes the application of machine learning decision support systems complex.
Confounding factors [16]	Catheters, devices, tubes, image quality, and patient position.

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
