# Peer review of "Chest X-ray in Emergency Radiology: What Artificial Intelligence Applications Are Available?"

_diagnostics, 2023, doi:10.3390/diagnostics13020216_

Round 1

Reviewer 1 Report

This paper gives a systematic review of the prospective application of AI in the field of chest X-ray Rays, which is of great reference value to researchers in related fields.

First of all, I think the author's paper is of high reference value, but before publication, the following two minors need to be revised:

1. The ref. [74] should be referenced on line 445 at Niehues et al.

     2. The conclusion of ref. [85] that china forbids the use of AI in the medical sphere” is one-sided. As we know, On September 22, 2022, the Regulation of Shanghai Municipality on Promoting the Development of Artificial Intelligence Industry was voted through. It is clearly mentioned that Shanghai will promote the application and innovation of artificial intelligence in the medical field.

Author Response

The authors thank the reviewer for the work and precious suggestions.

We modified the manuscript, performing all the required changes, as follows:

This paper gives a systematic review of the prospective application of AI in the field of chest X-ray Rays, which is of great reference value to researchers in related fields.

First of all, I think the author's paper is of high reference value, but before publication, the following two minors need to be revised:

1. The ref. [74] should be referenced on line 445 at Niehues et al.

We corrected it accordingly

     2. The conclusion of ref. [85] that “china forbids the use of AI in the medical sphere” is one-sided. As we know, On September 22, 2022, “the Regulation of Shanghai Municipality on Promoting the Development of Artificial Intelligence Industry“ was voted through. It is clearly mentioned that Shanghai will promote the application and innovation of artificial intelligence in the medical field.

Thank you for this observation. We corrected the text according to this

Thank you again

Best regards

The authors

Reviewer 2 Report

The overall structure of the paper is appropriate, and the manuscript was written quite well. Some comments are provided to improve the quality of this manuscript. In my opinion, this manuscript should be revised before accepting in Diagnostics.

1) Authors stated AI limitations in section 7, please add a description regarding the limitations and drawbacks of using chest X-ray examinations.

2) The title of the manuscript is not self-explanatory. I recommend to choose a better title for the manuscript.

3) Please consider adding description of the paper's structure at the end of the Introduction.

4) In lines 94 and 95 the text is not coherent between two paragraphs. Please add one or two sentences to make it coherent.

5) The caption of tables should be above them.

6) Some texts in Table 1 are unreadable, and it is out of page margins. Please consider reformat it.

7) Some of references in section 2 are outdated (Ref [21], [25]). Please replace them with new ones, and mention newer articles.

8) Section 5 (COVID-19) misses several recent important works that use Chest X-ray for detection of COVID-19. I highly recommend to add the following papers to the manuscript to cover more related works:

     [1] "Automated detection of COVID-19 cases from chest X-ray images using deep neural network and XGBoost", Radiography. 28 (2022) 732–738. https://doi.org/10.1016/j.radi.2022.03.011.

     [2] "COVID-19 detection in chest X-ray images using a new channel boosted CNN." Diagnostics 12.2 (2022): 267. https://doi.org/10.3390/diagnostics12020267.

     [3] "A Novel Framework Based on Deep Learning and ANOVA Feature Selection Method for Diagnosis of COVID-19 Cases from Chest X-Ray Images", Comput. Intell. Neurosci. 2022 (2022) 4694567. https://doi.org/10.1155/2022/4694567.

9) Please add a Table and compare different proposed methods and mention their advantages and drawbacks. Please see Table 4 in the following reference for an example:

     [1] "A Novel Framework Based on Deep Learning and ANOVA Feature Selection Method for Diagnosis of COVID-19 Cases from Chest X-Ray Images", Comput. Intell. Neurosci. 2022 (2022) 4694567. https://doi.org/10.1155/2022/4694567.

10) Please add some real-world applications of AI in Chest X-ray examination software to the manuscript. I recommend to check the published papers in Software Impacts (Elsevier) journal.

11) Some of references in section 5 are outdated (Ref [54], [56], [57], [59], [60]). Please consider replacing them with new ones, and mention newer articles such as:

     [1] "Diagnosis of COVID-19 Cases from Chest X-ray Images Using Deep Neural Network and LightGBM," in: 2022 Int. Conf. Mach. Vis. Image Process., 2022: pp. 1–7. https://doi.org/10.1109/MVIP53647.2022.9738760.

     [2] "Segmentation-Based Classification Deep Learning Model Embedded with Explainable AI for COVID-19 Detection in Chest X-ray Scans." Diagnostics 12.9 (2022): 2132. https://doi.org/10.3390/diagnostics12092132.

     [3] "Classification of COVID-19 in Chest X-ray Images Using Fusion of Deep Features and LightGBM," in: 2022 IEEE World AI IoT Congr., 2022: pp. 201–206. https://doi.org/10.1109/AIIoT54504.2022.9817375.

Author Response

The authors thank the reviewer for the work and precious suggestions.

We performed all the required changes, as follows:

The overall structure of the paper is appropriate, and the manuscript was written quite well. Some comments are provided to improve the quality of this manuscript. In my opinion, this manuscript should be revised before accepting in Diagnostics.

1) Authors stated AI limitations in section 7, please add a description regarding the limitations and drawbacks of using chest X-ray examinations.

Thank you for the suggestion, we added a paragraph on this topics

2) The title of the manuscript is not self-explanatory. I recommend to choose a better title for the manuscript.

We tried to reformulate the title, according to this suggestion

3) Please consider adding description of the paper's structure at the end of the Introduction.

Thank you for the suggestion. We added a paragraph at the end of the introduction

4) In lines 94 and 95 the text is not coherent between two paragraphs. Please add one or two sentences to make it coherent.

Thank you for this observation, we revised the section and corrected this specific part

5) The caption of tables should be above them.

We moves them above the tables

6) Some texts in Table 1 are unreadable, and it is out of page margins. Please consider reformat it.

We modified the tables to make them correctly readable.

7) Some of references in section 2 are outdated (Ref [21], [25]). Please replace them with new ones, and mention newer articles.

Thank you for the suggestions, we updated these references with updated ones

8) Section 5 (COVID-19) misses several recent important works that use Chest X-ray for detection of COVID-19. I highly recommend to add the following papers to the manuscript to cover more related works:

     [1] "Automated detection of COVID-19 cases from chest X-ray images using deep neural network and XGBoost", Radiography. 28 (2022) 732–738. https://doi.org/10.1016/j.radi.2022.03.011.

     [2] "COVID-19 detection in chest X-ray images using a new channel boosted CNN." Diagnostics 12.2 (2022): 267. https://doi.org/10.3390/diagnostics12020267.

     [3] "A Novel Framework Based on Deep Learning and ANOVA Feature Selection Method for Diagnosis of COVID-19 Cases from Chest X-Ray Images", Comput. Intell. Neurosci. 2022 (2022) 4694567. https://doi.org/10.1155/2022/4694567.

We modified and updated the section on COVID-19 pneumonia AI-based systems detection 

9) Please add a Table and compare different proposed methods and mention their advantages and drawbacks. Please see Table 4 in the following reference for an example:

     [1] "A Novel Framework Based on Deep Learning and ANOVA Feature Selection Method for Diagnosis of COVID-19 Cases from Chest X-Ray Images", Comput. Intell. Neurosci. 2022 (2022) 4694567. https://doi.org/10.1155/2022/4694567.

Thank you for the observation. We inserted a new table as required

10) Please add some real-world applications of AI in Chest X-ray examination software to the manuscript. I recommend to check the published papers in Software Impacts (Elsevier) journal.

Thank you for the suggestion.

We add examples and related images.

11) Some of references in section 5 are outdated (Ref [54], [56], [57], [59], [60]). Please consider replacing them with new ones, and mention newer articles such as:

     [1] "Diagnosis of COVID-19 Cases from Chest X-ray Images Using Deep Neural Network and LightGBM," in: 2022 Int. Conf. Mach. Vis. Image Process., 2022: pp. 1–7. https://doi.org/10.1109/MVIP53647.2022.9738760.

     [2] "Segmentation-Based Classification Deep Learning Model Embedded with Explainable AI for COVID-19 Detection in Chest X-ray Scans." Diagnostics 12.9 (2022): 2132. https://doi.org/10.3390/diagnostics12092132.

     [3] "Classification of COVID-19 in Chest X-ray Images Using Fusion of Deep Features and LightGBM," in: 2022 IEEE World AI IoT Congr., 2022: pp. 201–206. https://doi.org/10.1109/AIIoT54504.2022.9817375.

We updated our references to the suggested ones.

Thank you again

Best regards

The authors

Round 2

Reviewer 2 Report

In general, the authors have reflected my comments quite well and careful. In my opinion, the idea and experiments are qualified for publishing on Diagnostics.